# Grad-MobileNet: A Gradient-Based Unsupervised Learning Method for Laser Welding Surface Defect Classification

**DOI:** 10.3390/s23094563

**Published:** 2023-05-08

**Authors:** Sizhe Xiao, Zhenguo Liu, Zhihong Yan, Mingquan Wang

**Affiliations:** 1Beijing Research Institute of Automation for Machinery Industry, Beijing 100120, China; 2Faculty of Materials and Manufacturing, Beijing University of Technology, Beijing 100124, China; 3School of Software Engineering, Beijing Jiaotong University, Beijing 100044, China

**Keywords:** manufacture of power batteries, welding defect detection, unsupervised learning, gradient-based model

## Abstract

Deep learning technology has advanced rapidly and has started to be applied for the detection of welding defects. In the manufacturing process of power batteries for new energy vehicles, welding defects may occur due to the high directivity, convergence, and penetration of the laser beam. The accuracy of deep learning prediction relies heavily on big data, but balanced big data of welding defects is hard to acquire at the battery production site. In this paper, the authors construct a dataset named RIAM, which consists of images captured from an industrial environment for laser welding of power battery modules. RIAM contains four types of images: Normality, Lack of fusion, Surface porosity, and Scaled surface. The characteristics of RIAM are carefully considered in the application scenarios. Moreover, this paper proposes a gradient-based unsupervised model named Grad-MobileNet, which can be trained with only a few normal images and can extract the feature gradients of the input images. Welding defects can then be classified by the gradient distribution. This model is based on MobileNetV3, which is a lightweight convolutional neural network (CNN), and achieves 99% accuracy, which is higher than the accuracy expected from supervised learning.

## 1. Introduction

In the process of manufacturing power battery modules of new energy vehicles, generally a high-power laser oscillator welding process is used to weld the battery lugs, which are made of copper and aluminum. In the welding process, there are many causes of welding defects, such as fluctuations in welding parameters, oxidation on the surface of the aluminum pole column, inadequate gas protection, large busbar gaps, etc. Due to changes in welding conditions, many welding defects inevitably arise, such as porosity, slag, broken welding, blackening, etc. These defects have a serious impact on the safety of the battery and are very likely to cause battery pack fires. For the above welding defects, achieving “early diagnosis, early detection, early treatment” from the source of the process has become an urgent quality problem to be solved.

In recent years, defect detection methods based on computer vision and deep learning have received widespread attention, but there are still three challenges for researchers: Firstly, the lack of datasets is a common problem for the majority of researchers. The GDXray-weld dataset proposed by Mery D. et al. [1] dataset is widely used in the field of weld defect detection, but this dataset mainly collects internal weld defects and is in small quantities. No authoritative public dataset for new energy vehicle power battery welding surface defects has been published. In the actual production scenario, there are many inconvenient and time-consuming ways to collect and obtain a full range of datasets. Defect detection based on a small and unbalanced dataset has become a difficult and hot research point. Secondly, welding defect classification is mainly based on supervised methods. With the development of deep learning, supervised learning methods dominate in computer vision. As a member of the traditional manufacturing industry, researchers in the field of welding focus more on the types of defects to improve the process. Thus, supervised learning methods have a greater advantage. Last but not least, most unsupervised learning methods can only do binary classification. In the field of defect detection, the lack of defective samples has led researchers to explore methods that only use positive samples. Thus, the method can reconstruct and discriminate normal samples and is less capable in multi-categorization.

To address the aforementioned problems, this paper proposes a gradient-based unsupervised learning algorithm named Grad-MobileNet. This model has achieved good results in the classification of laser welding surface defects. In the course of the study, only positive samples are used for training; this algorithm can also identify positive and negative samples. Then, the negative samples can be classified as different defects.

Unsupervised learning accomplishes the classification task by describing the distribution of the input data. It requires many complex features to be designed manually, which traditional unsupervised learning algorithms apply to welding defect classification tasks. Further, a gradient-based method is mainly used in explaining the black-box model. In this paper, we apply it to the classification task. The region of interest of the input image can be obtained by observing the gradient of the input image while training the model. By means of gradients, this algorithm can get the features needed for unsupervised learning. Thus, it makes the unsupervised learning classification task easier and more efficient.

The contributions of this paper are as follows:This paper introduces a laser welding surface defect dataset named RIAM, which was collected from plants. The differences in features between this dataset and other publicly available datasets are analyzed. The photos were taken by vision sensors. According to ISO 6520, there are four categories in the dataset: Normality, Lack of fusion, Surface porosity, and Scaled surface.In order to cope with problems such as lack of datasets and poor model interpretability, this paper proposes a laser welding surface defect classification method, namely Grad-MobileNet, which is an unsupervised learning algorithm based on MobilenetV3. The algorithm not only shows high recognition accuracy, outperforming supervised-learning-based methods, but is simple and efficient to train. It is sufficient for automatic detection and classification of cell welding defects in production.In this article, we compare and analyze the differences between the industrial defects datasets and other publicly available datasets. A new unsupervised industrial defect classification method is provided for subsequent researchers.

The rest of the paper is organized as follows. Section 2 covers a review on the gradient-based method and unsupervised learning. Section 3 describes the dataset captured by visual sensors. Additionally, it analyzes the differences between the welding dataset and other publicly available datasets. The idea behind the algorithm’s design and the algorithm flowchart are presented in Section 4. Section 5 describes the experimental design and its results. Finally, Section 6 presents the conclusions of the paper and suggests some directions for future research.

## 2. Related Work

The idea behind this paper draws on recent work in CNN welding defect classification, gradient-based methods, and unsupervised learning algorithms.

**CNN welding defect classification:** As one of the most effective algorithms for classification tasks, convolutional neural networks are widely used for welding defect classification. Many previous studies [2,3,4,5,6] have achieved good results in the field of welding defects. Liu et al. [2] proposed a model based on VGG16 that enables high-precision identification of both porosity and cracks with a small dataset. Hou et al. [3] proposed a DCNN model and compared the effectiveness of three resampling methods to solve the defective-sample-imbalance problem. Agus Khumaidi et al. [5] used a CNN and Gaussian kernel to identify four different types of welding defects. Daniel Bacioiu et al. [6] used an HDR camera to capture weld images and constructed a classification model based on CNN and FCN. In addition to the changes in model structure, the researchers also used a number of other methods to improve model performance. Specifically Huang et al. [7] and Gu et al. [8] enhanced the unbalanced dataset and improved model accuracy by using generative adversarial networks (GANs). Further, Wang et al. [9] used edge computing to deploy a model on embedded devices and improved the detection speed while keeping the accuracy constant. V.A. Golodov et al. [10] used a digital detector array (DDA) method to preprocess features of the welding images and achieved 82.73% top1 and 96.76% top2 categorical accuracy. The above studies are all supervised learning approaches and were trained with a full range of labeled datasets.

**Gradient-based methods:** Gradient-based methods are generally used to explain black-box models due to the fact that CNNs are optimized using gradient descent. Consequently, there is no one using gradient-based methods in the field of welding defect classification. D. Smilkov et al. [11] proposed SmoothGrad to reduce the effect of noise until a clearer feature map is obtained. R. R. Selvaraju et al. [12] proposed Grad-CAM, which uses gradients to enhance CAM (Class Activation Mapping). It improves interpretability and allows gradients to be used for classification tasks. D. Omeiza et al. [13] proposed smooth grad-CAM++ which combines the methods from the two above techniques and produces more visually sharp maps with better localization of objects in the given input images.

**Unsupervised learning algorithms:** The main idea of unsupervised learning algorithms is to learn the data distribution. Further, they are trained without labels. Unsupervised learning algorithms for defect detection mainly use AE (Autoencoder) and GAN (Generative Adversarial Network). The following algorithms cover surface defects in fabrics, plastics, welds, etc. AE [14,15,16] can reconstruct the corresponding normal image based on the abnormal input. Further, abnormal images can be identified by the difference between the input and its reconstruction. Mei et al. [14] reconstructed image blocks using convolutional denoising autoencoder networks at different levels of Gaussian pyramid and integrated the detection results from these different resolution channels. The reconstruction residual of the training blocks was used as a direct pixel-level defect prediction indicator, and the residual maps generated by each channel were combined to produce the final detection result. This unsupervised, multimodal strategy can improve the robustness and accuracy of the model without human intervention.. Haselmann et al. [15] designed a deep convolutional neural network to perform anomaly detection on surface images in a block-wise manner. The method can generate a defect-free version of the completion region. By computing the pixel-level reconstruction error between the completion region and the query region, an anomaly score map is obtained, which can reveal surface defects. Kang et al. [16] proposed a method based on a deep denoising encoder, which can extract features from noisy images and reconstruct noise-free images. By comparing the original image and the reconstructed image, defect classification can be achieved. GAN [17,18,19] can learn the feature distribution of normal samples, so it can also be used to distinguish images with different distributions. Schlegl et al. [17] optimized AnoGAN and proposed f-AnoGAN to learn the manifold of normal anatomical variability, and they proposed an anomaly scoring scheme based on mapping from the image space to the latent space. The model can label anomalies and score image blocks, indicating their degree of matching with the learned distribution. Lai et al. [18] used a variational autoencoder (VAE) and generative adversarial network (GAN) to construct a surface texture pattern generator method, which can detect novelty according to the learned distribution. The experimental results on real industrial datasets show that the method can successfully generate surface texture patterns and effectively separate defects from normal regions by transforming images through the generator to the corresponding latent space. Hu et al. [19] proposed a novel fabric defect detection method based on a deep convolutional generative adversarial network (DCGAN). The authors added an encoder component to the standard DCGAN, which can reconstruct a given image so that the reconstructed image only retains normal texture without defects. The above two methods can only distinguish between normal and abnormal images and cannot do finer classification.

The fact that most existing welding defect detection methods need complete datasets does not match the real situation. The authors propose a new approach for defect classification based on production realities and dataset characteristics.

## 3. Welding Defect Dataset

In this section, we propose a novel welding defect dataset named RIAM. The dataset was collected from a power-cell production line. Afterwards, differences between different datasets are analyzed.

### 3.1. Introduction

The images in this paper were captured in an automotive power battery production plant. There are four classes of welding defects with resolution 256 × 256 pixels according to GB 6417 and ISO 6520. The whole dataset can be divided into four types: Normality, Lack of fusion, Surface porosity, and Scaled surface. Further, in this dataset, normality is a normal sample, and the other categories are abnormal samples. Figure 1 shows the schematic of laser welding surface defects.

### 3.2. Characterization

In the process of collecting defective datasets, normal images are more readily available than abnormal images. It is difficult for an algorithm to obtain a complete and comprehensive dataset at the time of training due to the limitations of professionals and the environment.

Industrial defect datasets differ from other publicly available datasets in terms of feature complexity. In the weld defect dataset, the weld and the defect are whole versus local and are low semantic versus high semantic. The coupling between them is relatively simple. In other datasets, the hierarchical relationship between low semantics and high semantics is complex and rich.

In the industrial defects datasets, defective areas are gradual and regular on the basis of pixel dots. As a result, normal images can be used as a reference for abnormal images. Theoretically, it is possible to classify normal and abnormal images as long as the difference in their distribution can be measured. Our proposed Grad-MobileNet can well conduct this task.

## 4. Algoritm and Structure

In this section, we detail the algorithm and architecture of our proposed Grad-MobileNet. Further, we discuss the classification strategy of the proposed model in detail.

### 4.1. Grad-MobileNet

Previous work has chosen the gradients of the convolutional layers or the predicted values. However, in order to better classify defects, this paper uses the gradient of the loss function in the testing phase.

Figure 2 shows the flowchart of the Grad-MobileNet. Grad-MobileNet is trained by using normal samples to obtain an overfit model. Then all kinds of images are fed into the overfit model to obtain $y_{pre}$, which can be given as:(1)$$y_{pre}=model\left(x\right).$$
where *x* denotes the input of the model and model(·) represents the mapping function from input to output. The gradient features can be calculated as:(2)$$Grad=\frac{\partial L(\theta ,y,y_{pre})}{\partial x}.$$
where L(·) is the cross-entropy function used as a loss function and θ denotes the trainable parameters. Since *x* consists of three features, which can be represented as $x=(x_{1},x_{2},x_{3})$, the L2 norm of gradients can be explicitly presented as:(3)$${∥Grad∥}_{2}=\sqrt{x_{1}^{2}+x_{2}^{2}+x_{3}^{2}}.$$

As shown in Equation (Equation 2), *y* is a one-dimensional matrix in which the values are all 0. Because of the derivative for input *x*, the dimension of Grad is the same as *x*. After obtaining the L2 norm of gradients by Equation (Equation 3), the gradient combines with the artificial landmark in order to reduce the randomness of the model.

It should be emphasized again that during the training phase the model can only be trained with normal samples until an overfit model is obtained. In the testing phase, after inputting the test set into the model, we need to set the *y* of the loss function in Equation (Equation 2) to 0. Finally, the gradient needed in this paper is obtained by deriving the loss function according to Equation (Equation 2).

### 4.2. MobileNetV3 Backbone

In the current study, deeper and more complex convolutional neural networks have better performance. However, this also brings the problem of slow operation with many parameters. So in this paper, the authors choose the lightweight network MobileNetV3 as the backbone.

MobileNetV3 was proposed by Andrew Howard et al. [20] from Google as a validation network, continuing the deep separable convolution of MobileNetV1 [21] and the inverse residual structure with linear bottleneck of MobilenetV2 [22], adding the squeeze and excitation based structure of MnasNet excitation structure, and also modifying the original swish activation function to h-swish for reducing the computational effort.

### 4.3. Artificial Landmark

AL means artificial landmark. Based on our experiment results, Grad-MobileNet has few errors randomly in the binary classification. As a result, this paper proposes a simple artificial landmark based on positive samples.

Figure 3 shows the grayscale image of the dataset and its histogram. The x-axis of the histogram represents the grayscale chart from dark to light. Because of the unsupervised method, this paper only designs the artificial landmark based on normal images. According to the normal histogram, the artificial landmark is designed from 0 to 50 in the x-axis of the histogram. This artificial landmark is too simple to classify all images, but it can reduce the randomness of the results.

### 4.4. Classification Rules

There are two situations for classification rules. From the perspective of images, one is black for binary classification and the other is for multiple classification. Binary classification uses one-class SVM to classify welding defects. Further, multiple classification use K-means. Figure 4 shows the classification strategy after the gradient input artificial landmark.

#### 4.4.1. One-Class SVM for Binary Classification

In this section, because the authors only have normal images, we choose one-class SVM for binary classification. One-class SVM is a variation of the SVM that can be used in an unsupervised setting for anomaly detection. A regular SVM finds a max-margin hyperplane that can differentiate normal images from abnormal images. Further, one-class SVM tries to make the hyperplane as close to the normal points as possible.

#### 4.4.2. K-Means for Multiple Classification

In the subsequent visualization experiment, high-dimensional images are easily separated by projection on a two-dimensional map based on the Grad-MobileNet. As a result, the authors use K-means to classify point clouds in a two-dimensional map.

## 5. Experiment

To demonstrate the feasibility of the Grad-MobileNet, this section shows the results of comparing the model with supervised learning model MobileNetV3 and InceptionNetV3. Because of the overfit model trained by the positive samples, all kinds of images can be regard as normal images and can be identified as the standard that can measure the type of defects. As a result, the experiment is divided into two stages. The first one is the binary classification that distinguishes between the positive and negative class. The second part is defect classification that identifies all types of defects from the negative class.

### 5.1. Metrics

For classification models, the performance evaluation metrics are generally measured by using four values of a confusion matrix, which are True Positive (TP), False Positive (FP), False Negative (FN), and True Negative (TN). Based on these four values, the metrics for evaluating the classification model can be calculated. In this paper, the authors choose accuracy rate to evaluate the model performance.

The accuracy rate refers to the ratio of the number of correctly classified samples to the total number of classifications, and it is the most commonly used evaluation index to measure the performance of classification models in general; it is calculated as follows:(4)$$ACC=\frac{TP+TN}{TP+TN+FP+FN}.$$

### 5.2. Experimental Results on RIAM

#### 5.2.1. Binary Classification

Binary classification is the preliminary and validation step of the experiment. It aims to demonstrate the feasibility of the experimental idea. Therefore, the authors only use Grad-MobileNet and one-class SVM for classification. Table 1 shows a highly accurate binary result. Due to the significant difference between positive and negative classes, the authors achieved good results. This result confirms that an overfit model can perform well for classification tasks. In the next chapter, it is applied to a finer classification task.

#### 5.2.2. Defect Classification

Table 2 compares the accuracy rates among Grad-MobileNet, MobileNetV3 and InceptionNetV3. Grad-MobileNet is trained with only 50 normal images, while MobileNetV3 and InceptionNetV3 hlsplit the dataset into training and test sets at a ratio of 7:3. In the testing phase, our model classifies the defects based on the new distribution identified by the gradient analysis since the types of defects are unknown beforehand. The new types discovered are numbered in order.

Table 2 shows that the accuracy rates of Grad-MobileNet are higher than those of MobileNetV3 and InceptionNetV3. And the bolded part represents the model with the highest accuracy rate among the three types of models. This indicates that the unsupervised learning model proposed in this paper has better performance on RIAM.

This section presents the result in the gray box. The number of classifications depends on experience and the results of dimensionality reduction. Therefore, no labeling is required for the images in the dataset.

### 5.3. Analysis of Results

To gain deeper insight into the model and image features, the authors randomly selected one hundred images of each defect type and fed them into the model to observe their distribution. Figure 5 shows the histogram of the gradient L2 norm. The horizontal axis of the graph represents the mean of the gradient L2 norm calculated from Equation (Equation 3). The vertical axis of the graph represents the frequency. Table 3 shows the mean and variance of the histogram. In Figure 5, the range of normal images is from 1.2 × $10^{-4}$ to 9.4 × $10^{-4}$; the range of surface-porosity images is from 6.2 × $10^{-5}$ to 5.3 × $10^{-4}$; the range of lack-of-penetration images is from 5.7 × $10^{-5}$ to 1.0 × $10^{-3}$; the range of scaled-surface images is from 7.0 × $10^{-5}$ to 3.2 × $10^{-4}$.

Figure 6 shows the gradient point cloud of each pixel computed from Equation (Equation 2). The three axes correspond to the three components of the gradient. The gradient only indicates the direction and magnitude of the pixel value change and does not have any physical meaning. Due to the overfitting of the model, the point clouds of different categories are hard to distinguish in the original space. Therefore, we cannot classify images based on Figure 5 and Figure 6 alone.

Figure 7 shows the classification result in a two-dimensional plane. Figure 7a is obtained by directly projecting Figure 6 onto a two-dimensional plane. Figure 7b,c are the clear classification results after applying T-NES. Since different random seeds in T-NES may lead to different outcomes, this paper presents two types of seed results. This also demonstrates the good performance in the pre-experiment.

## 6. Conclusions

This paper presents a novel method for the classification of welding defects without a labeled dataset. It demonstrates that gradient-based methods can be used not only to interpret black-box models but also to perform classification tasks. The criteria for evaluating the ultra-accurate prediction of defects are the classification accuracy and the consistency of the gradient distribution among the same defect type. Our method achieves 0.99 accuracy, which is higher than the accuracy of the supervised learning model. Analysis of the experiment shows that the gradient distribution of the same defect type is similar, while different defect types have distinct gradient distributions. Therefore, our method can capture the essential features of different defect types and distinguish them effectively.

The validity of our method is supported by the strong correlation between positive and negative samples in defect datasets. Our method uses this strong coupling relationship to measure the distance between different defect types and positive samples. This distance is represented by the gradient when optimizing the model.

The scientific novelty of our method lies in using gradient-based methods for unsupervised defect classification, which has not been explored before. Our method can overcome the limitations of pixel value methods and supervised learning methods, which require labeled data or prior knowledge. Our method can also handle unbalanced datasets, which are common in industrial scenarios.

The results of the experiment are based on a dataset named RIAM, which consists of images captured from an industrial environment for laser welding of power battery modules. RIAM contains four types of images: Normality, Lack of fusion, Surface porosity, and Scaled surface. The diagnostic equipment used in the experiment is a high-resolution camera mounted on a robotic arm.

The authors hope that our method can facilitate unsupervised learning research in defect classification and provide a new way of thinking for industrial applications. 

## Figures and Tables

**Figure 1 sensors-23-04563-f001:**
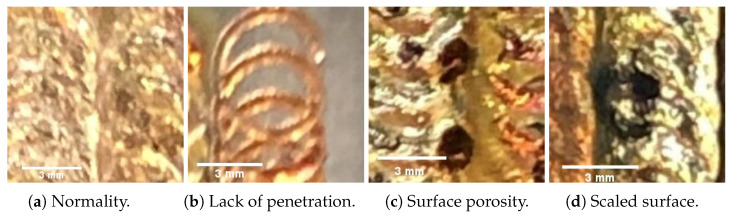
Laser welding defects.

**Figure 2 sensors-23-04563-f002:**
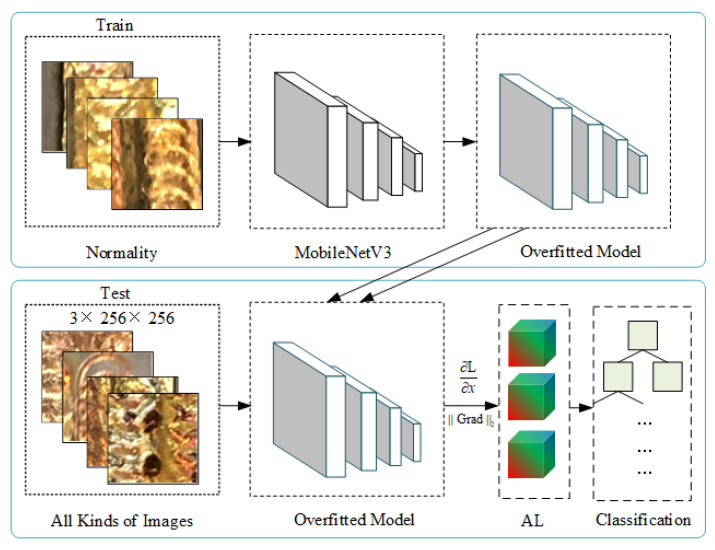
Flowchart of the Grad-MobileNet.

**Figure 3 sensors-23-04563-f003:**
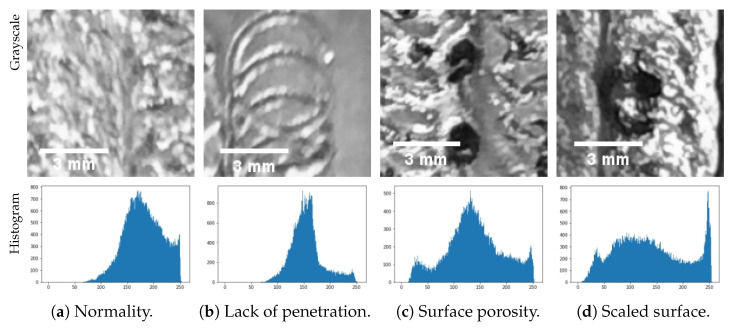
Grayscale images and their histogram of welding defects.

**Figure 4 sensors-23-04563-f004:**
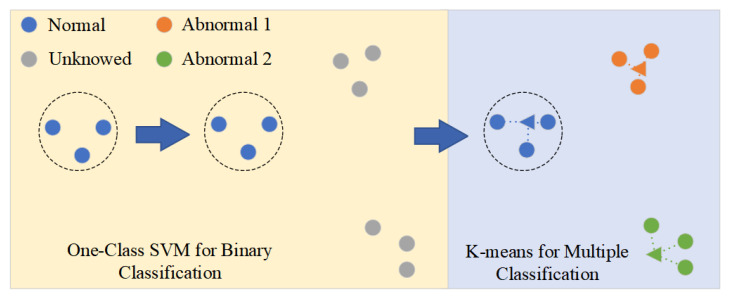
Classification rules of welding defects.

**Figure 5 sensors-23-04563-f005:**
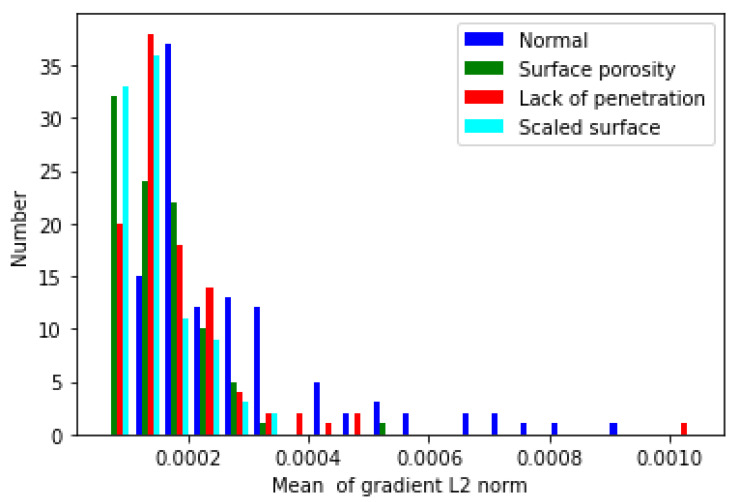
The histogram of gradient L2 norm.

**Figure 6 sensors-23-04563-f006:**
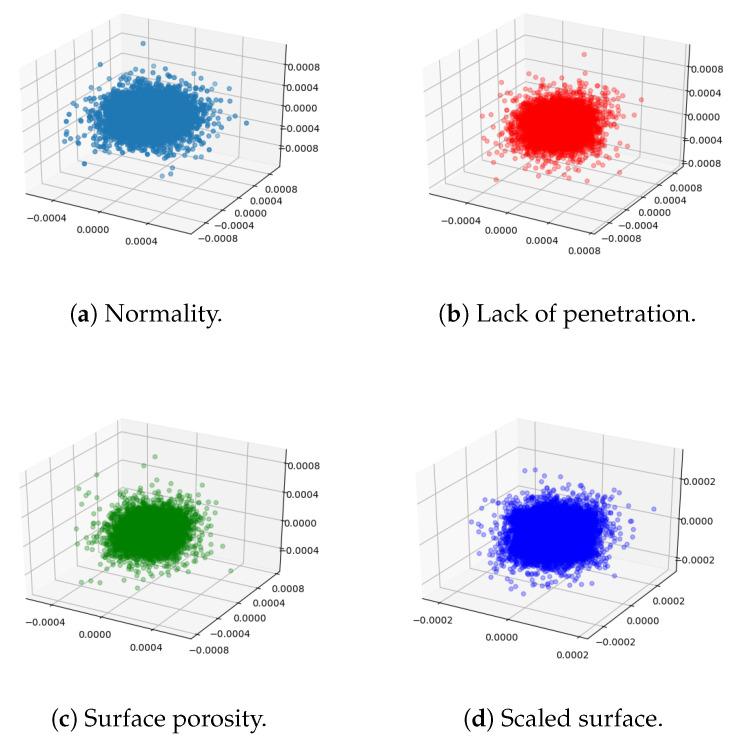
The gradient point clouds.

**Figure 7 sensors-23-04563-f007:**
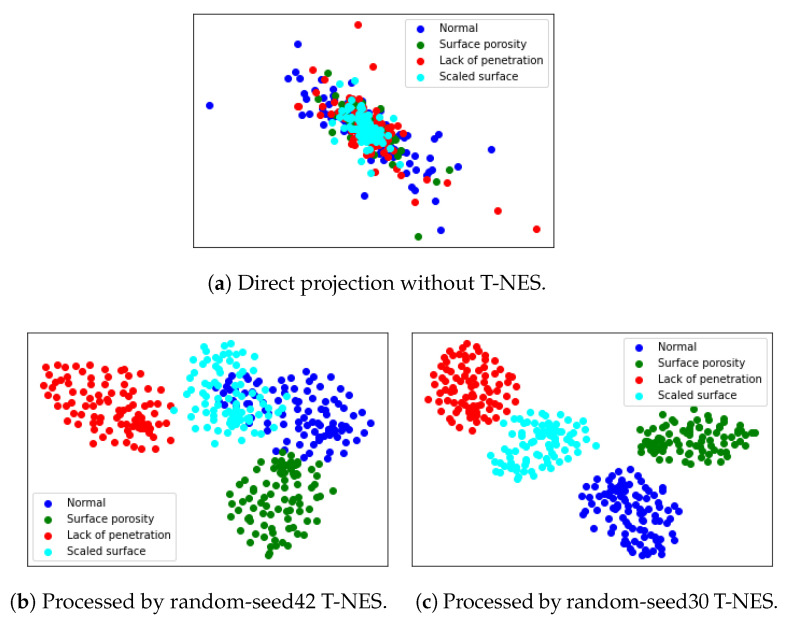
The gradient point clouds in a 2-dimensional map.

**Table 1 sensors-23-04563-t001:** Binary classification.

Image Type	Grad-MobileNet Accuracy
Normal Images	0.98
Abnormal Images	0.98

**Table 2 sensors-23-04563-t002:** Accuracy Comparison.

Image Type		Grad-MobileNet	MobileNetV3	InceptionNetV3
Normal Images		**1.000**	0.992	0.990
	Class 2 (surface porosity)	**0.991**	0.970	0.981
Abnormal Images	Class 3 (lack of penetration)	**0.991**	0.964	0.940
	Class 4 (scaled surface)	0.980	**0.987**	0.980
Sum		**1.000**	0.978	0.975

**Table 3 sensors-23-04563-t003:** The mean and variance of gradient features of all image types.

Image Type	Average Value	Variance
Normality	2.8 × $10^{-4}$	2.7 × $10^{-8}$
Surface Porosity	1.5 × $10^{-4}$	5.2 × $10^{-9}$
Lack of Penetration	1.8 × $10^{-4}$	1.4 × $10^{-8}$
Scaled Surface	1.4 × $10^{-4}$	3.2 × $10^{-9}$

## Data Availability

The data that supports the findings of this study are available from the correspoding author upon reasonable request.

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
