# Peer review of "Grad-MobileNet: A Gradient-Based Unsupervised Learning Method for Laser Welding Surface Defect Classification"

_sensors, 2023, doi:10.3390/s23094563_

Round 1

Reviewer 1 Report

This paper present a good work in the field of detecting welding defects. The authors proposed a method to detect welding defects, which depended on proposes Grad-MobileNet, whereas they used mobileNetV3 to classify welding defects and achieve promising accuracy 99%. On the other hand, the data used in this study not enough to depend on it, so try to use a large dataset in the future study plane.

Check the grammar and typing, e.g. line 189 delete the repeated word (the).

Author Response

Dear reviewers

Re: Manuscript ID: sensors-2296303 and Title: Grad-MobileNet: A Gradient-based Unsupervised Learning Method for Laser Welding Surface Defect Classification

Thank you for your letter and the reviewers’ comments concerning our manuscript entitled “Grad-MobileNet: A Gradient-based Unsupervised Learning Method for Laser Welding Surface Defect Classification”. Those comments are valuable and very helpful. We have read through comments carefully and have made corrections. Based on the instructions provided in your letter, we uploaded the file of the revised manuscript. Revisions in the text are shown using red highlight for additions, and strikethrough font for deletions. The responses to the reviewer's comments are marked in red and presented following.

We would love to thank you for allowing us to resubmit a revised copy of the manuscript and we highly appreciate your time and consideration. 

Sincerely.

Sizhe Xiao.

Reviewer 2 Report

1. According to GB 6417 and ISO 6520 Authors proposes 4 classes of welding defects:  Normality, Lack of fusion, Surface Porosity, and Scaled surface. However, in Figure 1 these classes are: Normal, Lack of penetration, Surface porosity and Scaled surface. The same names of classes should be used.

2. What is the authors' opinion on the resolution 256x256 px of the images shown in Figure 2? Should a higher one not be applied?

3. Normal and abnormal classes should be defined and described in the section 3. This information is included in Table 1. What is the 'abnormality criterion'? Are there other classes apart from Lack of penetration, Surface porosity and Scaled surface used in surface defect detection?

4. Grand-mobileNet algorithm should be described in detail. Two stages, i.e. Train and Test, should be presented separately and more precisely, taking into consideration steps presented in Figure 2.

Author Response

(The authors gave the same response as above.)

Reviewer 3 Report

1. The abstract contains general phrases. The authors need to specify which significant metal defects formed the database and what is the criterion for high-quality laser welding.

2. It is necessary to add keywords that characterize the research area.

3. In line 13, when describing, it is necessary to specify the brand of the material from which the battery module of new energy vehicles is made…

4. The report (lines 27 and 28) states that "No authoritative publicly available data set on weld surface defects has been published." It is necessary to explain:

- why does the text say that there is no data set on the defects of the seam surface? These data are sufficient, everything is described by scientists and there are pictures characterizing the shape of the surface, uniformity and color palette of defects depending on the material. Moreover, there is metrological diagnostic equipment with electronic interfaces that magnify and visualize the defect on the monitor. The authors need to justify why they believe that there is no data.

- Why are the data of surface defects accepted as a research program and a criterion for assessing the quality of a laser weld? All failures, breakdowns the decrease in durability and reliability of structures occurs due to progressively developing internal hidden defects in the structure of the material. It is the structure that bears the load, and technological defects reduce durability. But the internal defect is not determined by technical vision.

5. In line 54, it is required to list the types of defects that were analyzed by the program, which is taken for the quality of the laser seam during welding.

6. Explain what the vision sensor should see in the defect? What does it define?

7. On lines 80 and 81, when analyzing the forecasting models of the works of Liu, Hou, Agus Humaydi, etc., common words are presented. There is no specifics in the analysis of what exactly is interesting in these models and what they do not take into account. It is necessary to indicate the reliability of each diagnostic model in comparison in order to compare it with your methodology.

8. Lines 91 and 92 again contain common words. They talk about good results, what are they good results? What do you mean by good?

9. What in the article are controlled parameters for learning, and what are uncontrolled?

10 In lines 40 and 41 it is said that this article proposes a gradient-based learning algorithm that has no analogues in the world, called Grad-MobileNet, but further in the text from lines 95 to 100 are the names of scientists who the gradient method is actively and quite successfully used. This discrepancy causes a contradiction and requires clarification.

11. In line 106, it is required to specify and list what refers to abnormal input data.

12. In line 108 it is written that the image was reconstructed and synthesized. Explain why reconstruct the original shape and geometry of the part, if the defects dangerous for destruction have an internal structural character.

13. Explain what characterizes an ultra-precise and simply accurate self-coding network?

14. Lines 110 and 111 refer to defects in the surfaces of plastic parts, line 116 refers to fabric defects, and line 119 refers to welding defects. Explain which defects are investigated by the authors and which anomalies of which defects are considered in the algorithm using the gradient method?

15. Specify what characteristics of the dataset are you talking about in line 121? List it?

16. What devices produced the images in Figure 1?

17. How is the difference between normal and abnormal images measured, and in what units is it measured (line 145)?

18. Describe the methodology by which the defects were interpreted into the radio signal Figure 3.

19. Figures 6 and 7 require improving the quality of readability of indicators and numerical values.

20. After Figures 6 and 7, it is necessary to analyze the data and highlight the scientific significance.

21. The authors should present the experimental methodology in the form of a flowchart of the algorithm.

22. It is required to specify the boundary conditions of ultra-precise forecasting and the equation functions describing the reference weld and defective welds.

23. Improvement of the stylistic presentation of the research results is required.

24. Conclusions need to be completely redone.

It is necessary to clearly identify the criteria for evaluating ultra-accurate prediction of defects. Indicate the validity of the proposed method. Specify the scientific novelty and the results of the experiment. It is not clear whether the technique has been tested in production conditions?

It is required to specify which diagnostic equipment was used in the experiment.

25. The conclusions do not contain scientific research results. What is obtained, what is established, dependencies or a pattern. What you recommend. Is the hypothesis confirmed or not?

In general, the article needs to be finalized and, if comments are eliminated and recommendations are implemented, it can be published. It is necessary to strengthen the scientific novelty and clearly formulate the goal, hypothesis and result. After eliminating the comments and recommendations of the reviewer, the article can be published.

Author Response

(The authors gave the same response as above.)

Reviewer 4 Report

In this manuscript, author proposed a gradient-based unsupervised model named Grad-MobileNet, which just needs a few normal images to train and can achieves 99% accuracy. Also the authors compare the model with supervised learning MobileNetV3; the experimental results show that the model can achieve good results. The following are some suggestions for the revision of the manuscript.

1. Figure 1 and Figure 3 missing scale bar.

2. There are few comparative trials in the paper authors just compare the model with supervised learning MobileNetV3, i Suggest adding 1-2 algorithmic models for comparison.

3. This paper proposes a gradient-based unsupervised model named Grad- MobileNet, Where is this unsupervised approach reflected, please explain it in detail.

Author Response

(The authors gave the same response as above.)

Round 2

Reviewer 3 Report

The article is recommended for acceptance for publication in this form.